# Preliminary Investigation on Marine Radar Oil Spill Monitoring Method Using YOLO Model

Bo Li [1,2,†], Jin Xu [1,2,*], Xinxiang Pan [1,2,*], Rong Chen [1,2,†], Long Ma [1,2], Jianchuan Yin [1,†], Zhiqiang Liao [1], Lilin Chu [1], Zhiqiang Zhao [1,2], Jingjing Lian [3] and Haixia Wang [3]

1   Naval Architecture and Shipping College, Guangdong Ocean University, Zhanjiang 524091, China
2   Shenzhen Institute of Guangdong Ocean University, Shenzhen 518116, China
3   Navigation College, Dalian Maritime University, Dalian 116026, China
*   Correspondence: jinxu@gdou.edu.cn (J.X.); panxx@gdou.edu.cn (X.P.); Tel.: +86-158-4269-5866 (J.X.)
†   These authors contributed equally to this work.

**Abstract:** Due to the recent rapid growth of ocean oil development and transportation, the offshore oil spill risk accident probability has increased unevenly. The marine oil spill poses a great threat to the development of coastal cities. Therefore, effective and reliable technologies must be used to monitor oil spills to minimize disaster losses. Based on YOLO deep learning network, an automatic oil spill detection method was proposed. The experimental data preprocessing operations include noise reduction, gray adjustment, and local contrast enhancement. Then, real and synthetically generated marine radar oil spill images were used to make slice samples for training the model in the YOLOv5 network. The detection model can identify the effective oil spill monitoring region. Finally, an adaptive threshold was applied to extract the oil slicks in the effective oil spill monitoring regions. The YOLOv5 detection model generated had the advantage of high efficiency compared with existing methods. The offshore oil spill detection method proposed can support real-time and effective data for routine patrol inspection and accident emergency response.

**Keywords:** marine radar; oil spill; YOLO

## 1. Introduction

Oil spill disasters have tremendous adverse effects on the ecological environment, human health, and economic development of coastal cities. In June 2011, the offshore oil field *19-3* continued to leak into the Chinese Bohai Sea [1]. The oil spill was not completely cleaned off for 2 months, and 5500 square kilometers of seawater were polluted. On 5 April 2021, oil leaked from rig *V29* of the China National Offshore Oil Corporation (CNOOC). The sea around the oil rig continued to show obvious plumes on high-resolution satellite images. On 27 April of the same year, the Panamanian cargo ship *Sea Justice* collided with the Libyan oil tanker *M/V Symphony* in the southeast sea of Chaolian Island, Qingdao, China, resulting in oil spill from the damaged cargo hold and tanker [2].

Pollution caused by offshore oil leakage is extremely harmful. Effective and rapid oil spill monitoring and early warning actions are of great significance to disaster prevention and emergency response. Satellite oil spill monitoring technology is widely studied [3]. Yu et al. extracted oil spill data based on region growing, edge detection, and adaptive threshold using the EnviSat radar images in Dalian, China on 17 July 2010 and the Chinese Bohai Sea, on 11 June 2011 [4]. Song et al. put forward a robust active contour model (ACM) for segmenting oil films in Synthetic Aperture Radar (SAR) images based on a statistical energy function that combines the smooth function, the level set function, and the constant approximating of the true signal from the corresponding object [5]. Cao et al. established an automatic oil spill classification model with relatively few training samples through the Active Learning (AL) method, and extracted oil spill regions and suspected oil spill targets in RADARSAT images [6]. Song et al. adopted a Gaussian smoothing operator to

further weaken the speckles in the polarimetric SAR images and make the oil spill detection contour evolution of ACM more stable based on Lee filter operator [7]. Xu et al. carried out oil spill classification in RADARSAT remote sensing images based on machine learning methods [8]. Chen et al. proposed an oil spill segmentation method based on the SAR imaging mechanism. This method used the synchronization of SAR images to establish a framework for oil spill segmentation [9]. Airborne oil spill monitoring technology has developed rapidly due to its advantages of convenience. Liu et al. advanced the oil film classification method in Airborne Visible Infrared Imaging Spectrometer (AVIRIS) remote sensing images combining spectral index-based band selection (SIs) and one-dimensional convolution neural network [10]. Chen and Lu realized an oil spill identification method in Unmanned Aerial Vehicle (UAV) and aircraft images based on feature selection and machine learning [11].

Marine radar oil spill monitoring technology has the advantage of accompanying ship clean-up actions, and has developed rapidly in recent years. Ships, islands, and buoys produce high radar echo intensity and show highlight image features in the marine radar images. The oil spill targets can absorb or smooth the radar electromagnetic wave, which appears relatively dark in the sea wave echo regions of the marine radar images. The oil film information in the marine radar images can be extracted by using the relatively dark image feature. Axelson proved that marine radar could detect oil films within 1 km according to the evaluation experiment in 1974 [12]. In September 1987, the Canadian Environment Canada Minerals Management Service conducted an experiment of detecting oil spills with X-band shipborne navigation radar during a joint cruise in Nova Scotia, Canada, which verified the marine radar capabilities of oil spill tracking [13]. Afterwards, no progressive marine radar oil spill monitoring technology achievements were publicly released in a long time. Until 2010, researchers from Dalian Maritime University and Guangdong Ocean University have published positive marine radar oil spill monitoring and warning successes used marine radar images collected in the Chinese Dalian 7.16 oil spill accident [14].

Marine radar oil spill monitoring technology is mainly divided into three aspects: image preprocessing, effective oil spill monitoring region extraction, and oil film segmentation. Marine radar image preprocessing includes noise reduction, gray adjustment, and contrast enhancement [15]. Effective oil spill monitoring region extraction methods mainly include gray distribution matrix [16] and texture feature classification [17]. Oil film segmentation incorporates the threshold method [18], ACM [19] and fuzzy cluster analysis [20]. Deep learning has little application in marine radar oil spill monitoring technology. A convolutional neural network (CNN) architecture was used here to train limited image samples of and quickly extract effective oil spill monitoring regions. Among CNNs, You Only Look Once (YOLO) is a fast speed target detection model, which detects target locations in one go but maintains enough accuracy. Therefore, YOLO architecture is used in many fields, such as face payment, autopilot, and ship monitoring [21,22]. The existing marine radar oil spill monitoring technology consumes extensive computing time, and its real-time performance needs to be improved. The deep learning network occupies a large amount of memory during training. However, the target detection efficiency is extremely high. Therefore, this paper proposes using deep learning network to train the oil spill detection model for satisfying the needs of real-time monitoring. The deep learning network selected here for effective oil spill monitoring region detection is YOLOv5. YOLOv5 is a fast network which can support oil spill real-time detection. The oil spill segmentation method used here is local adaptive threshold.

The rest of this paper is structured as follows: Section 2 provides an overview about the raw data and methods. Section 3 presents the experimental results. Section 4 compares the performance with relevant research methods about the oil spill extraction. The conclusions are summarized in Section 5.

## 2. Materials and Methods

### 2.1. Materials

The experimental data are the original marine radar remote sensing images shown in Figure 1. They were collected on 21 July 2010 by the Sperry Marine radar system during the cruise monitoring mission of Dalian 7.16 oil spill accident governance action on the teaching-training ship *Yukun* belonging to Dalian Maritime University. The Sperry Marine radar system is used to monitor and record the wave clutter signals. The wave clutter signals received from the radar transceiver is directly connected to the computer processing system, and the image is displayed by the monitor after processing. With the rotation of the radar antenna, the radar system can digitize and store radar images of the sea surface. The radar adopts X-band single horizontal polarization and records 28–45 images per minute. The radar detection range can be controlled by adjusting the pulse width. In order to increase the image details of the oil films, the experimental data detection range is 0.75 nm, and the size of raw images is 1024 × 1024 pixels. Table 1 shows the performance characteristics of marine radar equipment.

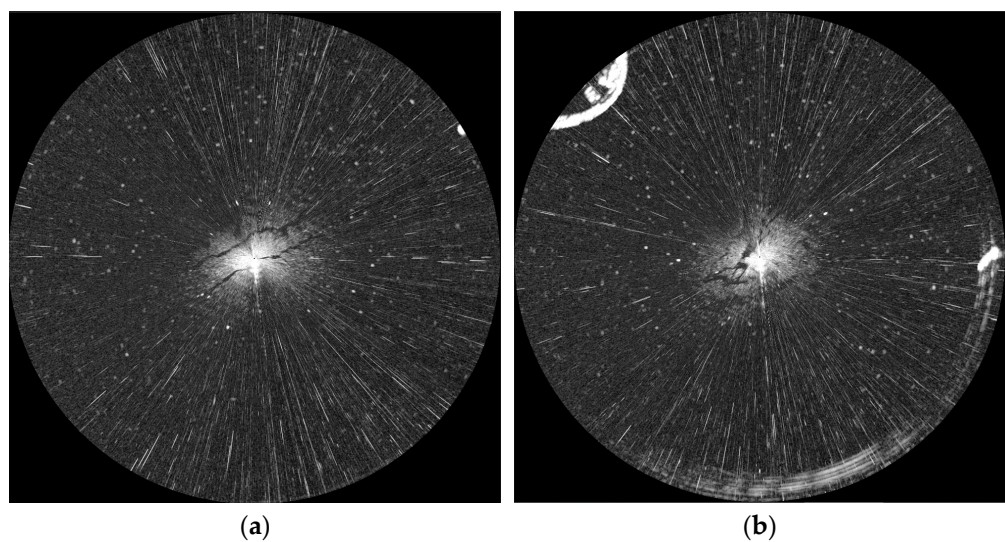

**Figure 1.** Marine radar remote sensing images of the Chinese Dalian 7.16 oil spill accident taken on 21 July 2010 at: (**a**) 23:19:34; (**b**) 23:21:24.

**Table 1.** The parameters of marine radar device.

| Performance Name | Technical Indicator |
|---|---|
| Operating frequency | X-band |
| Optional detection distance (nm) | 0.5/0.75/1.5/3/6/12 |
| Antenna type | Waveguide split antenna |
| Polarization mode | Horizontal polarization |
| Horizontal monitoring angle (°) | 360 |
| Rotation rate | 28–45 r/m |
| Antenna size (ft) | 8 |
| Pulse repetition frequency (Hz) | 3000/1800/785 |
| Pulse width (ns) | 50/250/750 |

### 2.2. Data Preprocessing

In practical application, the marine radar image adopts the Polar coordinate system with the position of ship as the center, and the azimuth and distance to describe the position of target. The wave echo data decreases with distance. So, the original marine radar image is converted from the Polar coordinate system to the Cartesian coordinate system with the azimuth as the horizontal axis and the distance as the vertical axis, which is convenient for

the overall adjustment of the image gray level. The coordinate conversion formula from Polar coordinate system to Cartesian coordinate system is:

$$\begin{cases} x = \rho \cos \theta \\ y = \rho \sin \theta \end{cases} \tag{1}$$

where $x$ and $y$ are the abscissa and ordinate of the Cartesian coordinate system, respectively, and $\rho$, $\theta$ are the distance and orientation of Polar coordinate system, respectively. The coordinate conversion formula from Cartesian coordinate system to Polar coordinate system is:

$$\begin{cases} \rho = \sqrt{x^2 + y^2} \\ \theta = \arctan\left(\frac{y}{x}\right) \end{cases} \tag{2}$$

The transformation of coordinate system is shown in Figure 2.

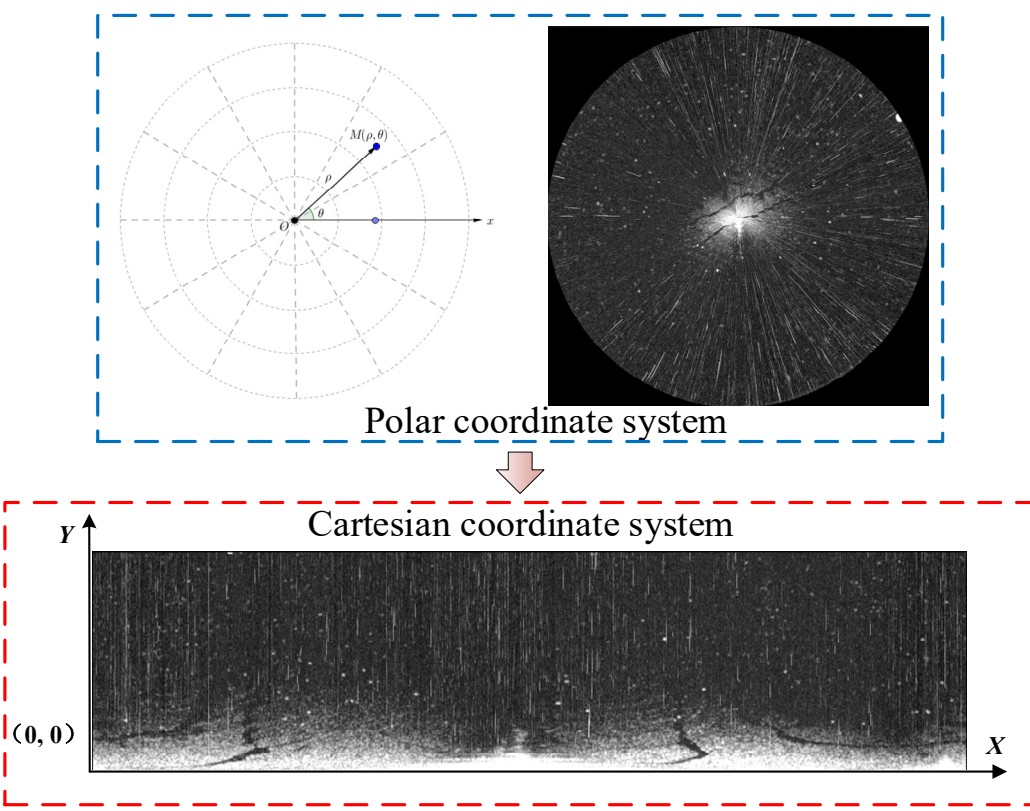

**Figure 2.** The transformation of the coordinate system.

The image preprocessing adopted the marine radar image processing method described in literature [19], as shown in Figure 3. Figure 1a was used to introduce the main preprocessed images as shown in Figure 4. Firstly, the marine radar image in Cartesian coordinate system was processed with Laplace operator and convolution. After that, Otsu threshold segmentation method was used to extract the co-frequency interference noises in the original marine radar image. Subsequently, the mean filter was used to suppress the co-frequency interference noises, as shown in Figure 4a. Secondly, gray threshold and isolated target area threshold were used to extract speckle noises in the image. Then, the median filter was used to smooth speckle noises to get noise reduction image, as shown in Figure 4b. Thirdly, gray correction matrix was generated to adjust the overall gray distribution of noise reduction image, as shown in Figure 4c. Finally, the local contrast enhancement model was used to enhance the contrast inside and outside the oil films, as shown in Figure 4d.

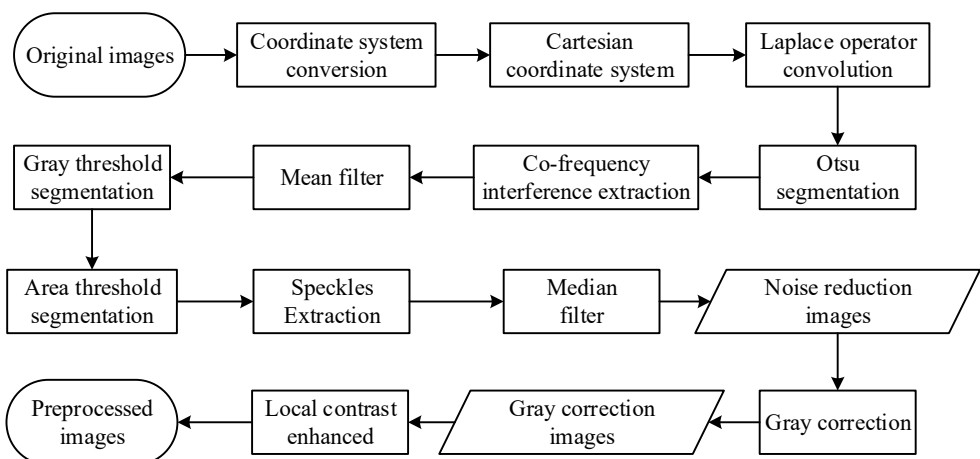

**Figure 3.** Data preprocessing scheme.

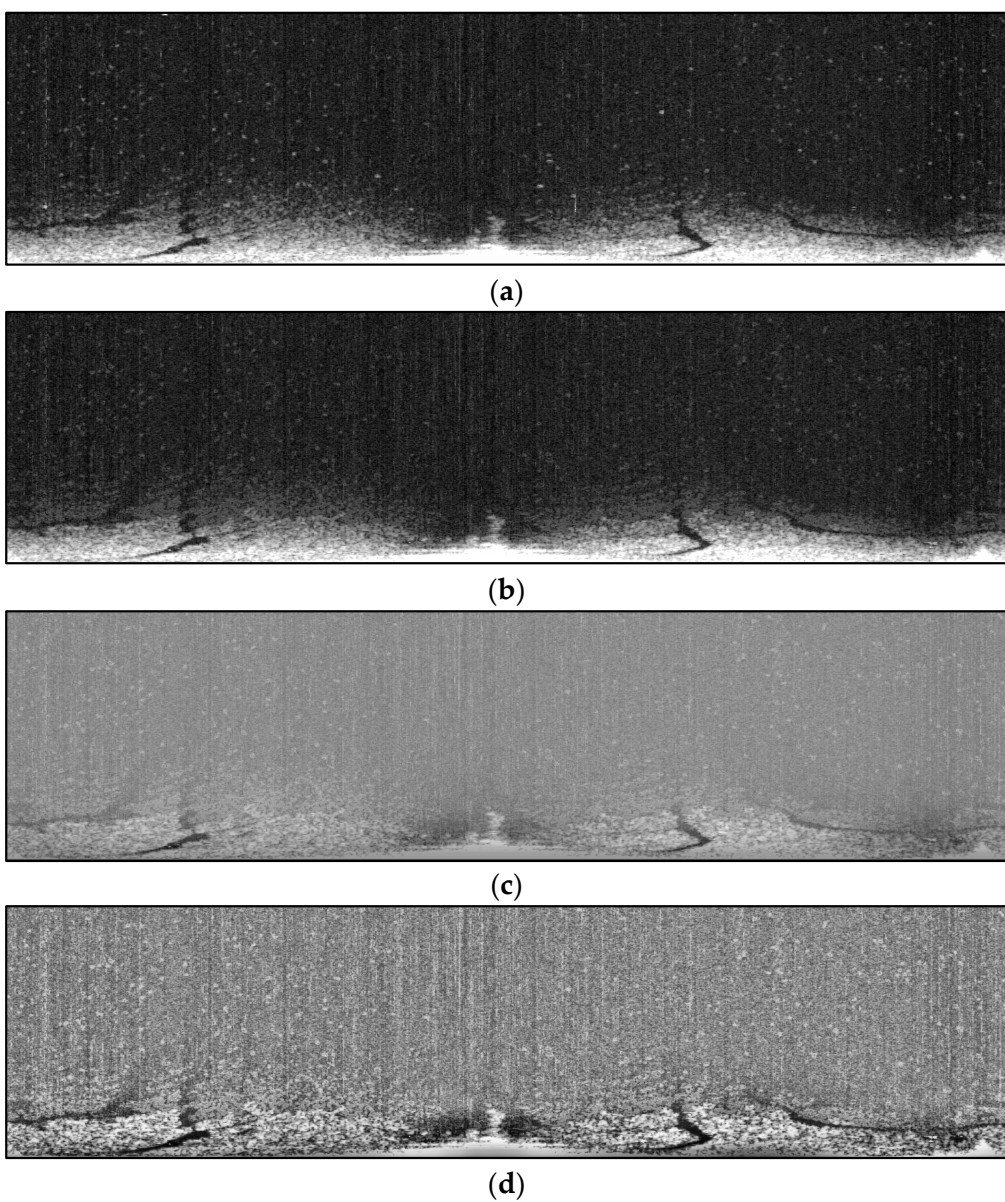

**Figure 4.** The preprocess. (**a**) Co-frequency interference noise suppression; (**b**) Speckle noise reduction; (**c**) Gray correction image; (**d**) Local contrast enhanced image.

### 2.3. Image Labeling

Oil film regions need to be manually labeled in the marine radar images because the YOLO model is a supervised learning network. The radar preprocessed image size in the Cartesian coordinate system is 512 × 2048 pixels. The preprocessed images containing oil films were cut into 512 × 512 pixels sub-images on average for training. In the real original data, there were 49 images with oil spills and 52 images without oil spills. So, 400 sub-images were generated. Due to the limited real oil spill samples, 100 marine radar images of different sea conditions were selected, and 50 of them were added with simulated oil films to generate syntactical images containing oil spills, as shown in Figure 5. Therefore, 200 positive sub-images and 200 corresponding negative sub-images were added and generated. Therefore, the total number of real and synthetic dataset was 800. Since the number of sub-images did not reach the desired scale, the proportion of training set was increased to 80% in order to obtain a model with good performance. The verification set and test set account for 10% and 10%, respectively. So, the numbers of sub-images in the training set, validation set, and test set were 640, 80, 80, respectively. The oil film regions in all samples were manually labeled in LableImg image label tool (Windows version 1.8.6, Heartex, San Francisco, CA, USA) developed by Python 3.9, as shown in Figure 6.

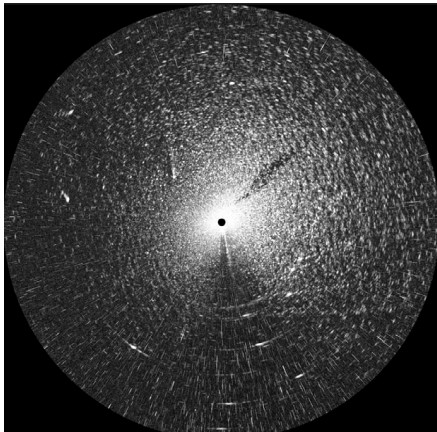

**Figure 5.** Synthetically generated marine radar oil spill image sample.

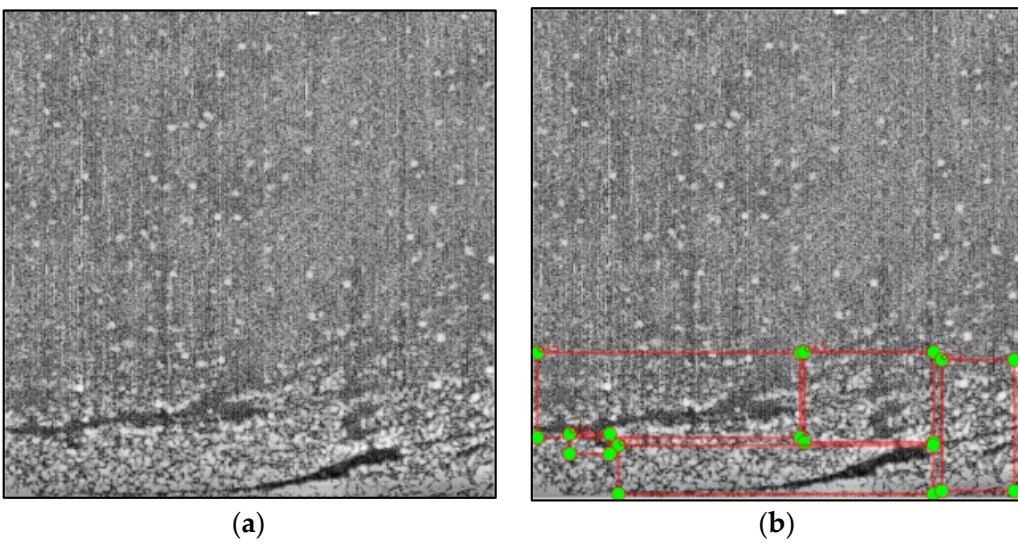

(**a**)                    (**b**)

**Figure 6.** Oil film labelling in the marine radar preprocessed image. (**a**) Preprocessed image tile sample; (**b**) Oil spill labels.

### 2.4. The YOLO Model

The spaceborne and airborne oil spill detection based on deep learning methods has become the mainstream technology. However, marine radar oil spill detection using deep learning methods is seldom. Deep learning technology can extract oil film features automatically in remote sensing data and avoid complex algorithm designs. According to the implementation process, deep learning technology can be divided into two-stage and single-stage [23]. The two-stage method, e.g., faster Region based Convolutional Neural Network (R-CNN), mask R-CNN, includes object region proposal and object classification. Due to the limitation of computer memory consumption and communication cost, the two-stage method is not as preferred as the single-order method [24]. The single-stage method, e.g., Single Shot Multi-Box Detector (SSD), YOLO, Fully Convolutional One-Stage Object Detection (FCOS), detects target without object region proposal. This type can be used in real-time dynamic monitoring because of its accurate and low consumption, which is a significant demand for oil spill clean-up and governance. Therefore, the one-stage method YOLO was used here for marine radar oil spill detection.

In the YOLO series, the YOLOv5 was one of the most popularly used object detection models [25–27]. The YOLOv5 was used to detect oil film regions in the marine radar preprocessed images. Its model architecture is depicted in Figure 7. The backbone plays the role of high-dimensional image feature extraction. The focus layer in backbone uses slicing operation to split the high-resolution image into multiple low-resolution feature images, and try to retain effective information while reducing the feature dimension. SPPF module in backbone realizes feather map fusion of local and global features. The neck part takes New CSP-PAN structure, which produces multi-scale low-dimensional image features. The function of the head outputs the target prediction boxes of the YOLO model. The YOLOv5s takes CSP-Darknet 53 as the backbone network and the path aggregation network as the neck, which has the fastest processing speed due to relatively few layers and nodes [21]. The YOLOv5s model was used here to establish the oil film grid identification model for reducing system occupation. The input image size was set to $64 \times 64$ pixels. This setting compressed the image to improve the operation efficiency. To effectively improve the utilization of video memory and prevent the continuous oscillation of loss function, the batch size was set to 16. In order to prevent an over-fitting of the model, the number of training epochs was set to 200. So as to reduce the model misjudgment and border superimposition, the confidence threshold was set to 0.275, and the intersection-over-union threshold of the model was set to 0.425. The hide labels parameter is set to true for improving the visibility of the oil film regions after detection. The best weight model after training completion is selected to detect new marine radar image based on the variation in box loss is:

$$Box\ loss = 1 - \frac{A \cap B}{A \cup B} \tag{3}$$

where $A \cap B$ is the intersection region of the manually labeled box and the predicted boundary box, and $A \cup B$ is the union region of the above two.

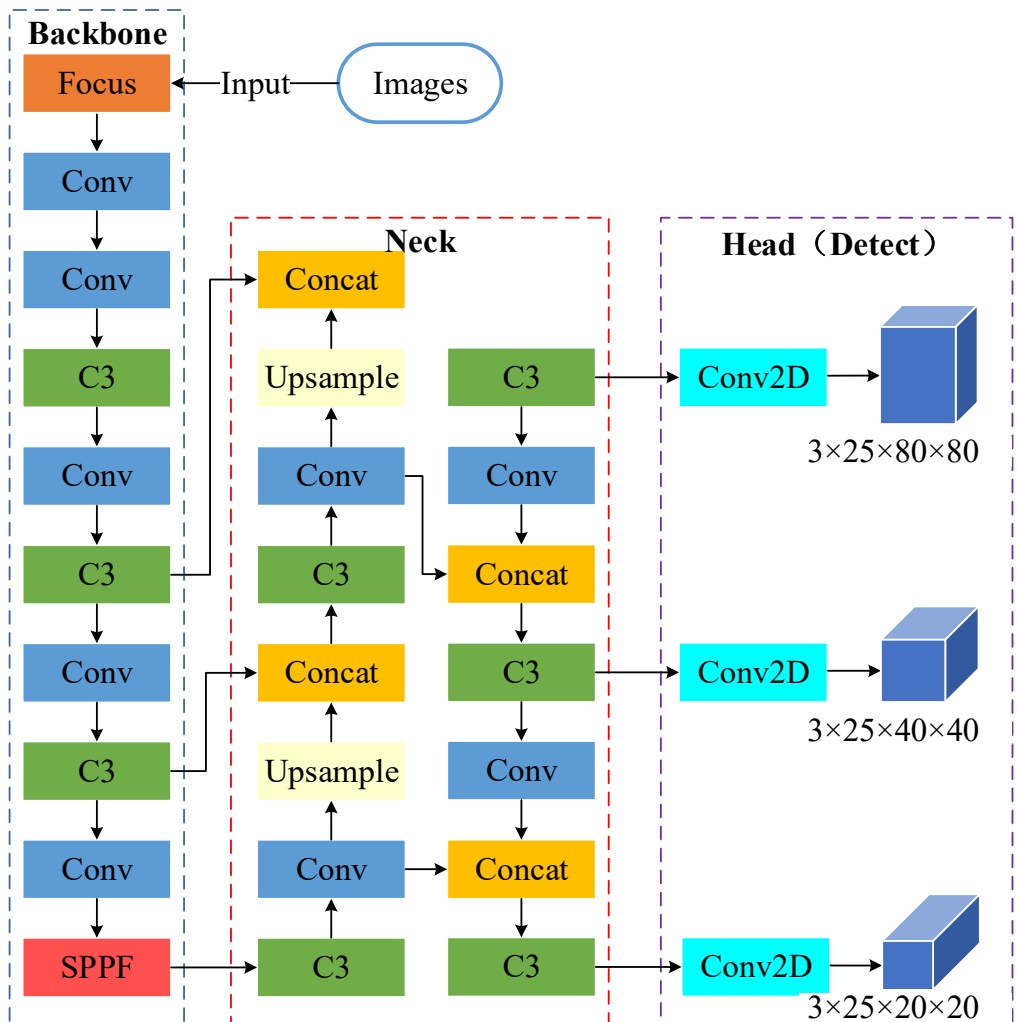

**Figure 7.** The architecture of YOLOv5 network.

*2.5. Adaptive Threshold*

Niblack [28] put forward a local image segmentation threshold as:

$$T = m + ks \tag{4}$$

where $m$ is the mean value, $k$ is a user-defined negative parameter value, $s$ is the standard deviation value. Sauvola and Pietikäinen [29] improved the threshold for document image binarization as:

$$T = m\left[1 + k\left(\frac{s}{R} - 1\right)\right] \tag{5}$$

where $k$ takes positive values, $R$ is the dynamic range value of standard deviation. Xu et al. [15] and Chen et al. [30] used the Sauvola threshold to realize the oil film segmentation in marine radar images. The Saulova threshold with the sliding local window is used to preliminarily extract the oil films in marine radar images. The parameter $k$ took 0.5. The sliding local window size was $64 \times 64$ pixels.

*2.6. Experimental Platform*

The toolbox used here is YOLOv5-master, built on PyTorch's deep learning library. The experimental platform is shown in Table 2.

**Table 2.** Software and hardware platforms.

| Platform | Name and Version |
| --- | --- |
| Operating System | Windows 10 |
| CPU | Intel(R) Core(TM) i7-10700F-2.9G |
| Graphics card | Nvidia Geforce GTX 1650-4G video memory |
| CUDA | 11.6 |
| CUDNN | 8.4 |
| Python | 3.9 |
| Pytorch | 1.13 |

## 3. Results

### 3.1. Curve Analysis of YOLO Model Training

The box loss variation during YOLOv5s training is shown in Figure 8. At the beginning, the box loss values of training set and prediction set descended swiftly. After 150 training epochs, the box loss curve of the verification set tended to stabilize. But the box loss curve of the training set kept on declining ditheringly. It indicated that the model began to over-fit after 150 training epochs. Therefore, the best detection model can be obtained after 150 training epochs. The box loss was around 0.10. The precise–confidence curve and recall–confidence were shown in Figures 9 and 10. It can be seen that the greater the confidence, the more accurate the oil spill detection in Figure 9. When the confidence threshold reached 0.708, the precise value was 100%. As can be seen from Figure 10, the lower the confidence level, the more comprehensive the oil spill detection results.

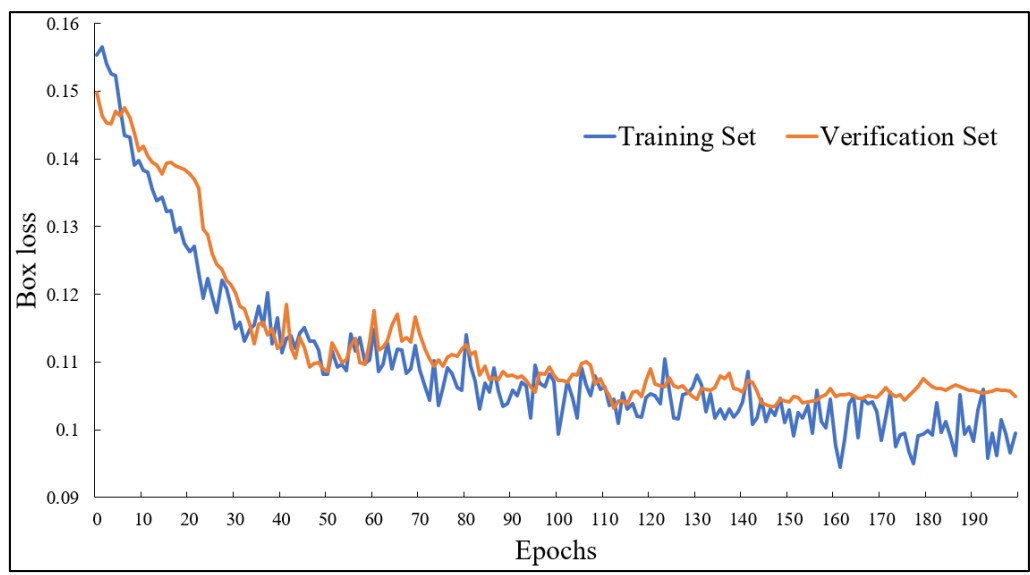

**Figure 8.** Variation of box loss.

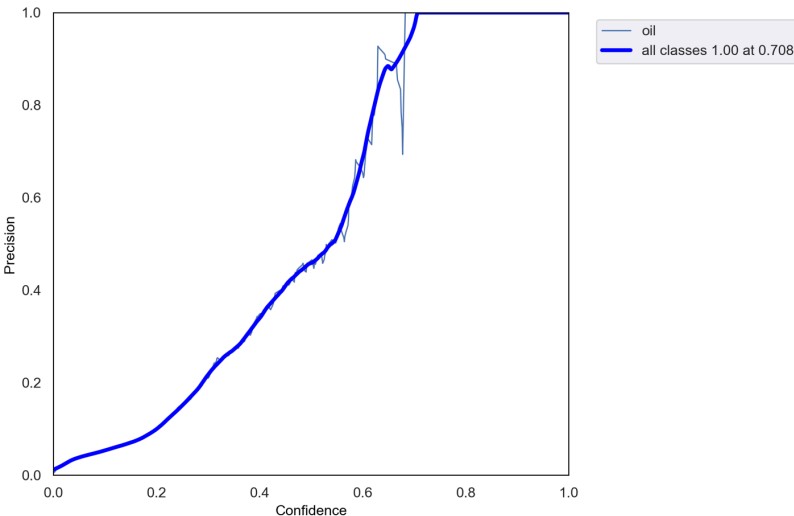

**Figure 9.** The precise–confidence curve.

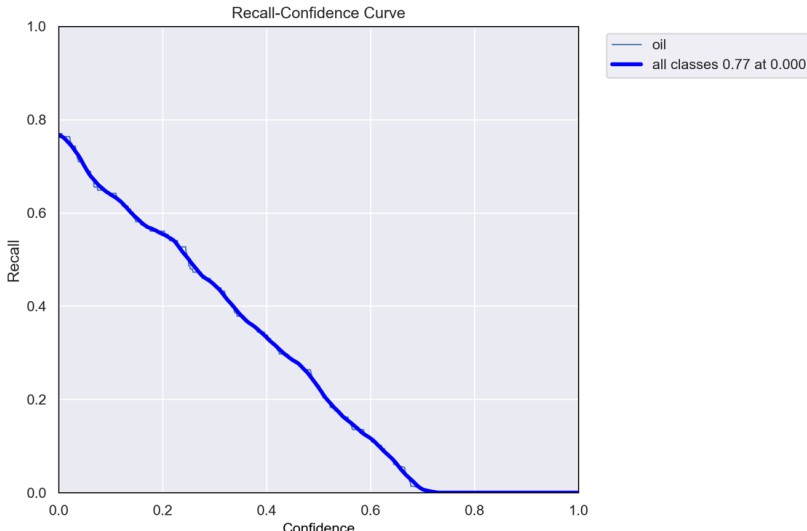

**Figure 10.** The recall–confidence curve.

### 3.2. Oil Spill Preliminary Identification

The trained YOLO model was used to mark the oil film regions on the preprocessed marine radar images. The detection speed is 1.3 ms per image. The effective oil spill monitoring regions were obtained in Figure 11b. Saulova threshold method was applied to segment the preprocessed images (Figure 4d), as shown in Figure 11c. Then, the Saulova threshold segmentation results were screened in the effective oil spill monitoring regions to obtain preliminary oil film segmentation as shown in Figure 11d.

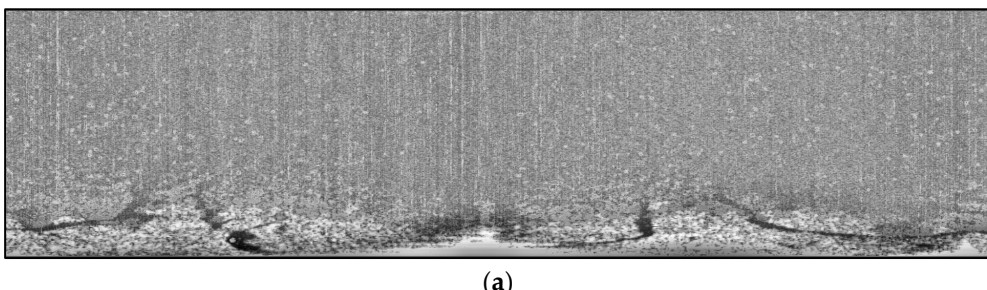

(**a**)

**Figure 11.** *Cont.*

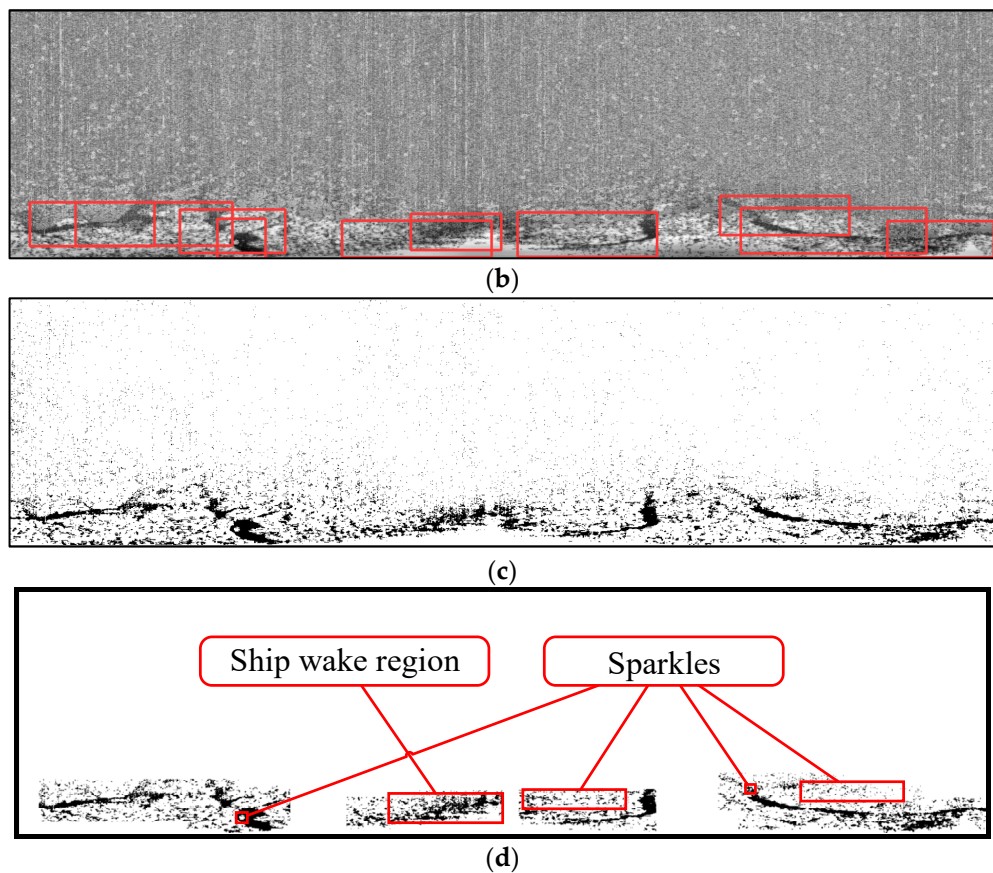

**Figure 11.** Oil spill preliminary identification. (**a**) The preprocessed marine radar image of Figure 1b; (**b**) The trained YOLO model detection result; (**c**) Saulova threshold segmentation; (**d**) Oil film preliminary segmentation.

### 3.3. Oil Spill Precise Identification

The process of oil spill precise identification is shown in Figure 12. In addition to the banded oil films, the preliminary segmentation image contains a large number of speckles inside and outside the oil films and ship wake regions, as shown in Figure 11d. The speckles and ship wakes need to be removed before extracting the oil films. At the beginning, blocks of 'Speckles inside the oil films' and 'Remove' in Figure 12 resulted in white spots removed in black banded oil films in Figure 13a. The isolated white spots inside the oil films were removed in the preliminary oil film segmentation image according to the pixel area threshold '100'. After that, blocks of 'Image inversion', 'Speckles outside the oil films' and 'delete' in Figure 12 resulted in white spots deleted outside the white banded oil films in Figure 13b. The binarization image was inverted, and the isolated white spots outside the oil film targets were deleted according to the pixel area threshold '200'. The ship wakes were also cut off and the precise oil film segmentation result was obtained in a Cartesian coordinate system. Then, blocks of 'Precise oil film segmentation', 'Merge' and 'Preprocessed image' in Figure 12 resulted in the red oil films marked in the noise reduction image (Figure 4b) in Figure 13c. The oil films were superimposed on the noise reduction image and marked in red. At last, blocks of 'Coordinate transformation' in Figure 12 resulted in the final results in Figure 13d,e. The oil film identification images were transformed into the Polar coordinate system from the Cartesian coordinate system.

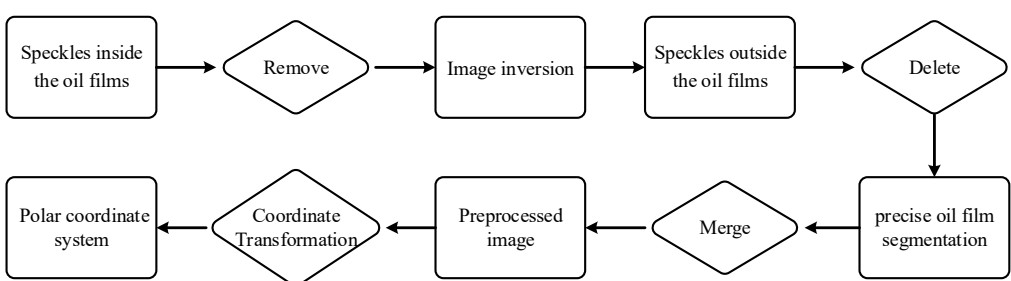

**Figure 12.** The process of oil spills' precise identification.

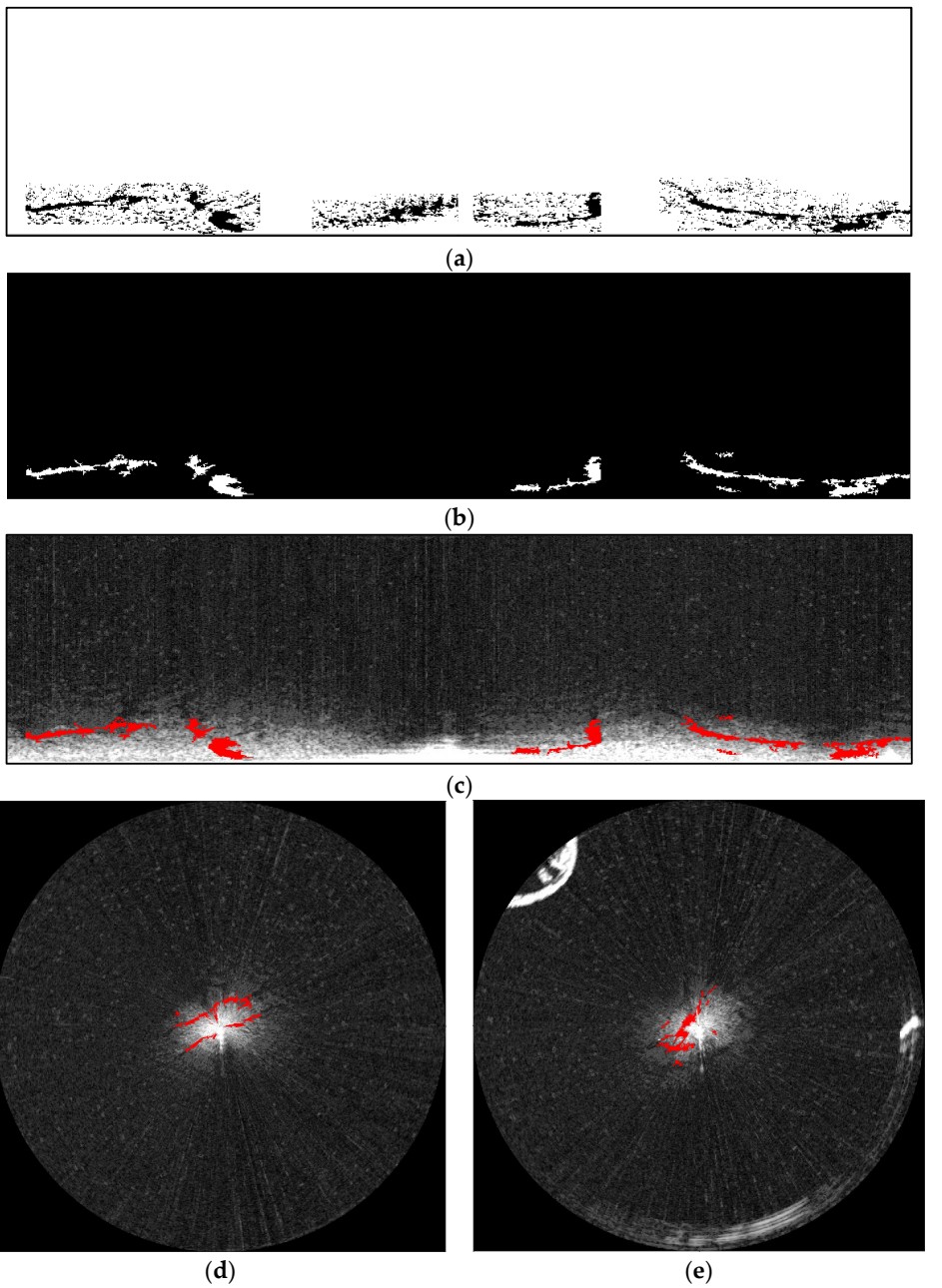

**Figure 13.** Oil spill precise identification. (**a**) The block noises were removed for the first time; (**b**) The block noises were removed for the second time in the flipped image; (**c**) Oil spill identification results in the Cartesian coordinate system. The red color targets were oil films, while (**d**,**e**) were oil spill identification results of Figure 1a,b in polar coordinate system, respectively.

## 4. Discussion

### 4.1. Oil Spill Target Identification Influence of Ship Wake and Long-Range Detection

In the noise reduction image, the generated wave echo regions were concentrated around the ship. There was no wave echo information in the image exceeding 0.375 nm range. Strong target neighborhood features in the long range seemed to be similar to oil films, which was easy to form false discrimination results, as shown in Figure 14a. In the contrast enhanced image, the ship wake region in the short range was prone to false target discrimination results because the regional features were similar to the oil films, as shown in Figure 14b. Therefore, this paper proposed the establishment of the oil spill remote undetectable region and the ship wake misjudgment region for screening automatically false positive identification targets, as shown in Figure 14c. However, the cost might be to delete some real oil film targets in ship wake region.

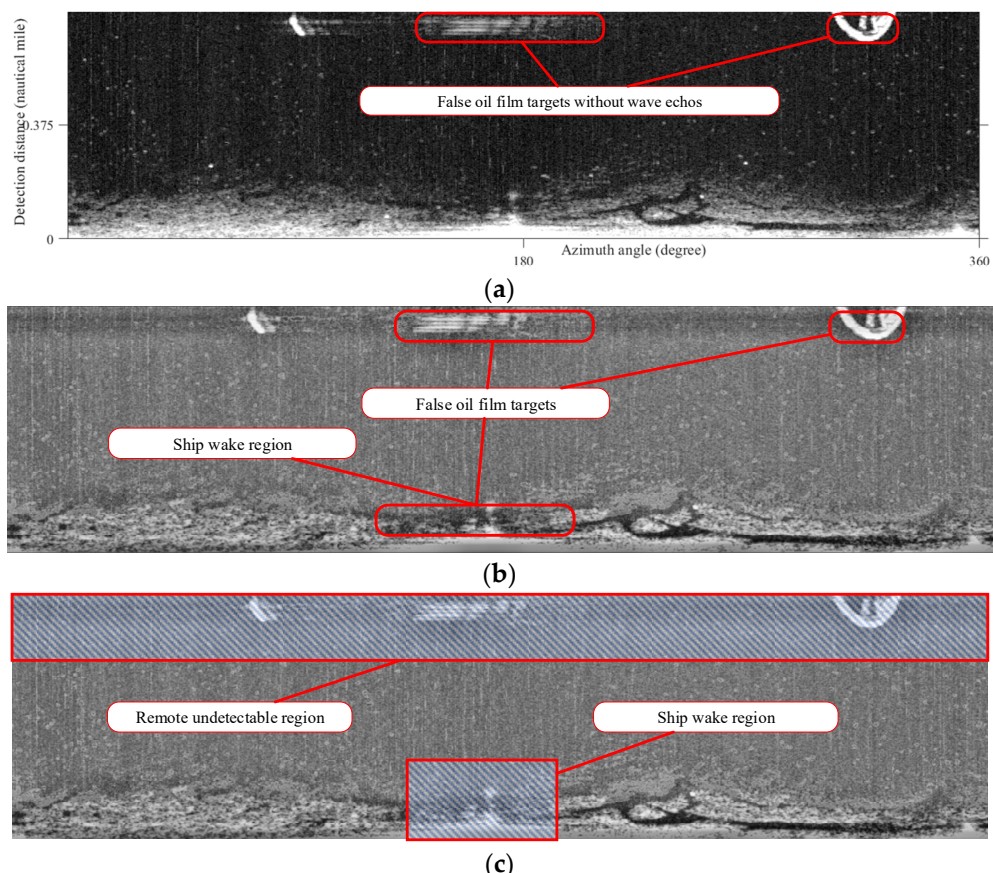

**Figure 14.** Oil spill misidentification conditions. (**a**) Misidentifications were generated in long detection range without no wave echo information in the noise reduction image; (**b**) Misidentifications were generated in ship wake region in the contrast enhanced image; (**c**) The oil spill remote undetectable region and the ship wake misjudgment region were established.

### 4.2. Effect of Image Local Contrast Enhancement

The contrast between the oil films and the neighborhoods was reduced after the overall image gray correction. The gray correction images were used for slicing and training, and the effective oil film monitoring regions were detected in Figure 15a. When the effective oil spill monitoring regions were extracted, continuous oil film cracked as shown in Figure 15b. The gray correction smoothed the regions without wave information, but reduced the contrast between the inside and outside of the oil films, leading to missing detection. Slicing and training with contrast enhanced images can effectively avoid oil film detection fracture, as shown in Figure 15c,d.

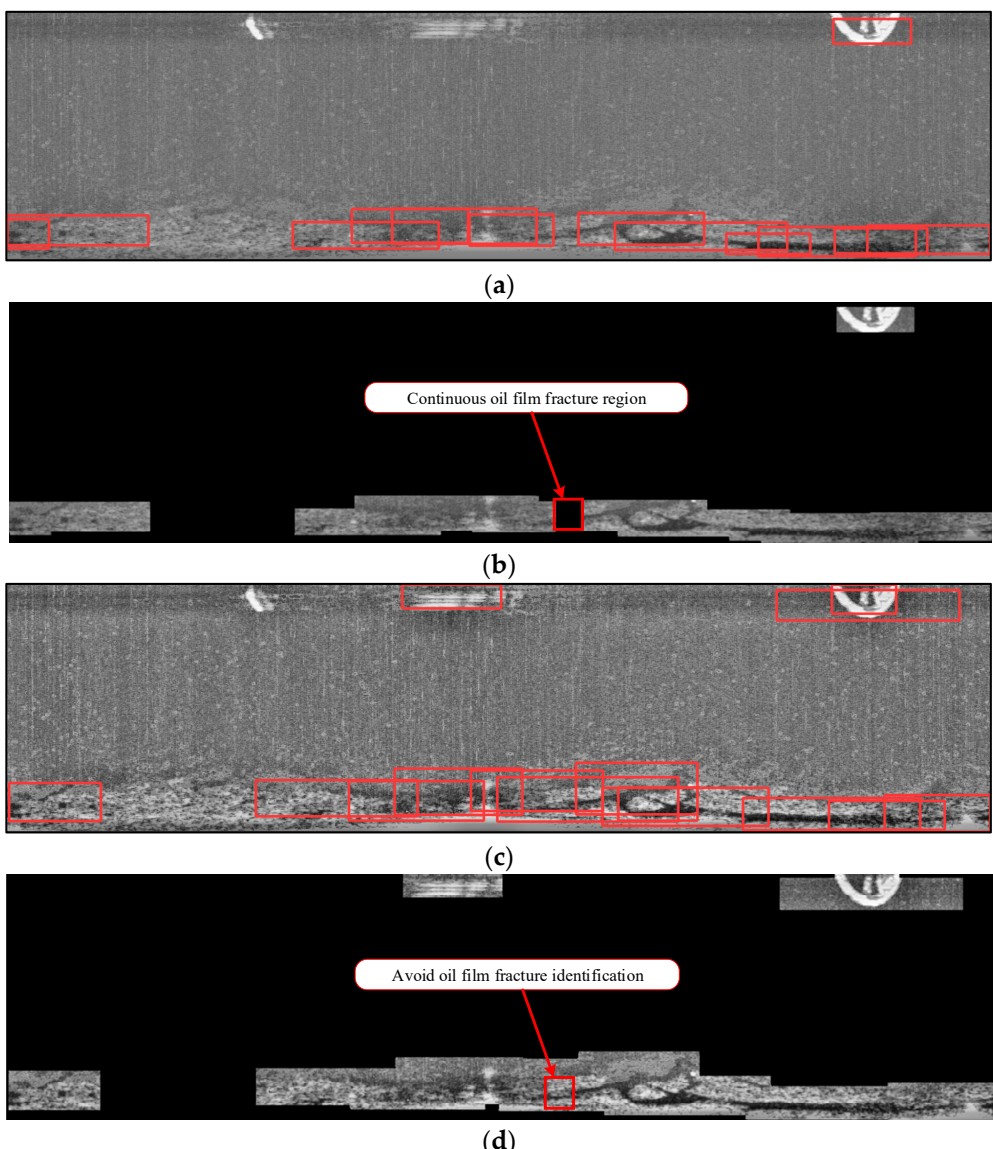

**Figure 15.** The influence of contrast enhancement on oil film detection. (**a**) Oil spill detection using trained model of gray correction images; (**b**) The effective oil spill regions extraction of (**a**); (**c**) Oil spill detection using trained model of contrast enhancement images; (**d**) The effective oil spill regions extraction of (**c**).

*4.3. Comparison of Existing Oil Spill Identification Methods*

Marine radar effective oil spill monitoring region was first proposed by Liu et al. [16]. They employed one $20 \times 160$ matrix to calculate the mean value in the local window of each pixel in the noise reduction image in order to create the gray distribution matrix, as shown in Figure 16a. Then, the gray distribution matrix was segmented (Figure 16b) according to the gray threshold to obtain the effective oil spill monitoring region (called Method 1 here), as shown in Figure 16c. The manual threshold selection of the segmentation gray distribution matrix is the key to extracting the effective oil spill monitoring region. In recent years, the combination of image texture features and machine learning (called Method 2 here) has become the mainstream method for extracting effective oil spill monitoring regions, such as local binary mode (LBP) and K-Means [17], or Gray level co-occurrence matrix (GLCM) and Support Vector Machine (SVM) [20], as shown in Figure 17.

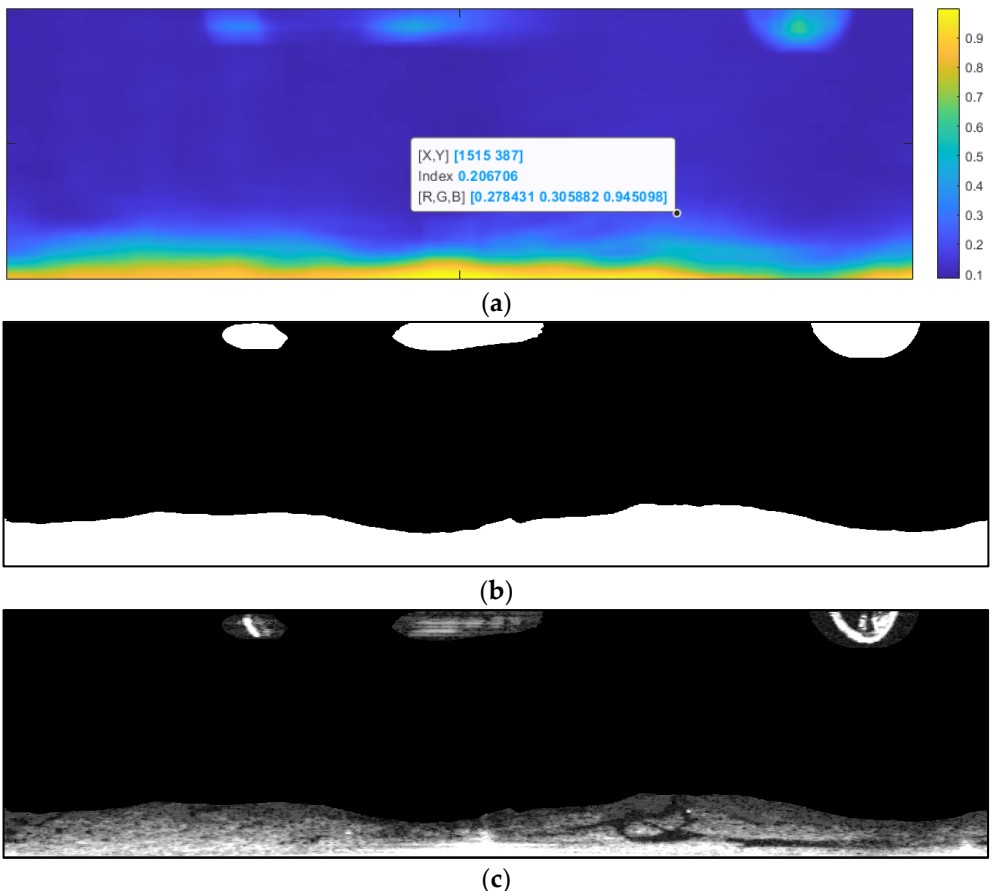

**Figure 16.** Effective monitoring regions extraction with gray distribution matrix. (**a**) Gray distribution matrix; (**b**) Threshold segmentation; (**c**) Effective monitoring region extraction.

Methods 1 and 2 can obtain reasonable oil spill monitoring regions. Method 1 took 0.85 s due to a simple calculation of the gray distribution matrix. It is more efficient than texture feature and machine learning methods. However, Method 1 involved manual threshold segmentation. If the segmentation threshold was low, more invalid monitoring regions would be obtained. If the segmentation threshold was high, real effective monitoring regions would be reduced. Method 2 can be implemented automatically, and the calculation efficiency depended on the feature extraction and machine learning classification process. LBP feature calculation is uncomplicated, so the consumption time of Figure 17b was 4.19 s. The feature calculation of the GLCM is relatively complex. Although the efficiency of Figure 17d was improved by slicing scheme instead of sliding window, the overall consumption time was still 14.34 s. Our method took a long time in model training, but the actual detection process only consumed 1.3 ms, which performed much better than Methods 1 and 2. Therefore, the accident emergency command action can obtain real-time data support by cooperating with the oil spill deep learning detection method.

Since the approximate effective oil spill monitoring regions were obtained in Figures 16b and 17a, Saulova threshold was only used to identify the oil films with Method 2, as shown in Figure 18. The oil spill identification results obtained by method 2 were similar to that of Figure 13e. However, the effective oil spill monitoring regions in Figure 17b,d were essentially the effective regions of sea wave echo, not the location of oil films. Therefore, some suspected fake oil films appeared.

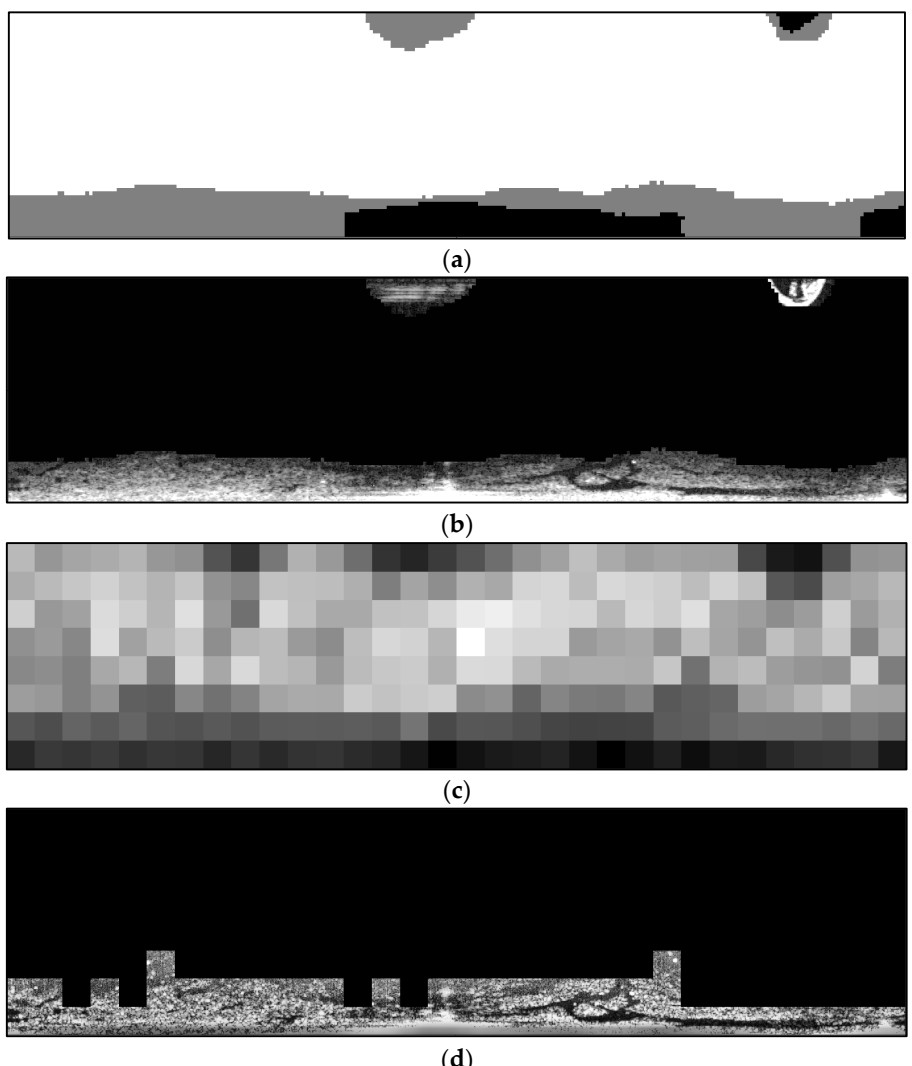

**Figure 17.** The effective monitoring region extraction with texture features and machine learning. (**a**) LBP feature and K-means classification result: The sliding window size was 128 × 128, the classification number is 3; (**b**) The effective monitoring region extraction with LBP feature and K-means; (**c**) Contrast feature of GLCM: The tile window size was 64 × 64; (**d**) The effective monitoring region extraction with GLCM and SVM.

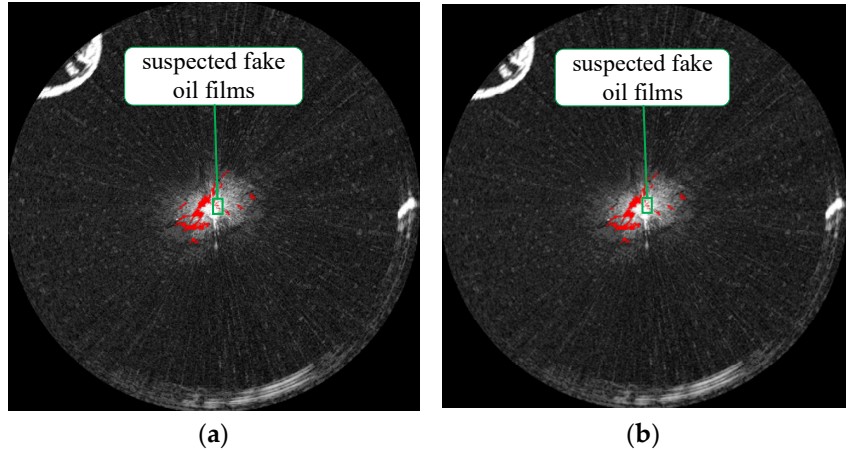

**Figure 18.** Oil identification of image texture features and machine learning. (**a**) LBP feature and K-means classification result; (**b**) Contrast feature of GLCM and SVM.

### 4.4. The Influence of the Different Settings in YOLO Deep Learning Network

The oil film detection results of YOLO Deep Learning Network with different settings are shown in Figure 19. When the number of epochs was set to 100, the trained detection model cannot accurately obtain the position of the oil films, indicating that the LOSS curve of the model was not stable, as shown in Figure 19a. When the number of epochs was set to 300, the result was similar to the number of epochs set to 200, which showed that the fitting of the LOSS curve had been stable before the epochs reached 200, as shown in Figures 19b and 11b. When the batch sizes were set to 8, 16 and 32, the detection performances of the trained oil spill detection model were similar, as shown in Figure 19c,d. A small amount of oil films in Figure 19c was not identified, and the performance effect was inferior to that in Figure 11b. With the increase of batch size, the memory consumption and time of training also increased. Therefore, it was recommended to set the batch size to 16 here. If the image size was set to $512 \times 512$, the detection model can hardly locate the oil films. When the image size was set to $256 \times 256$, there were some invalid identification regions, as shown in Figure 19e. When the image size was set to $128 \times 128$, a few oil films were not identified, as shown in Figure 19f. Therefore, the recommended image size was $64 \times 64$ here.

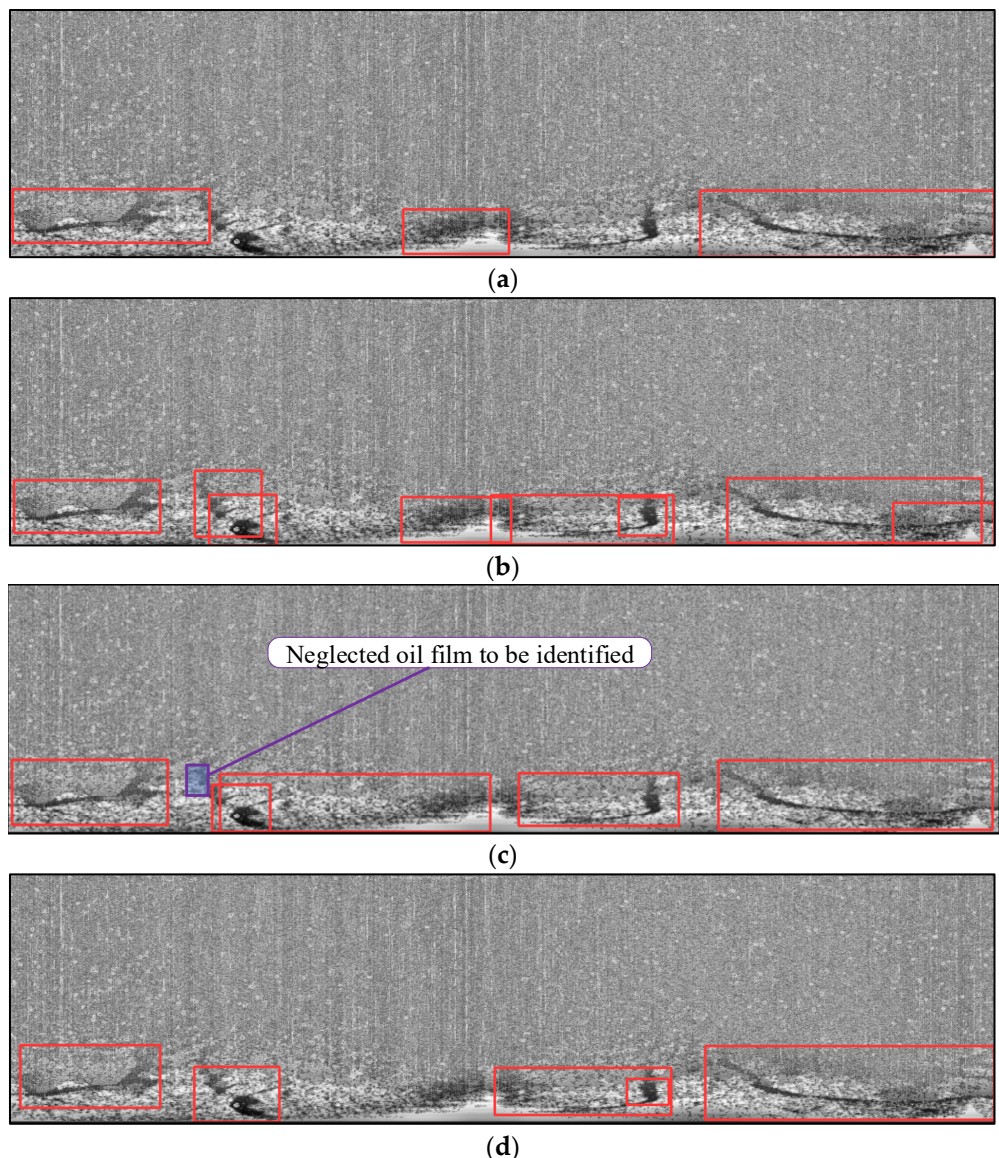

**Figure 19.** *Cont.*

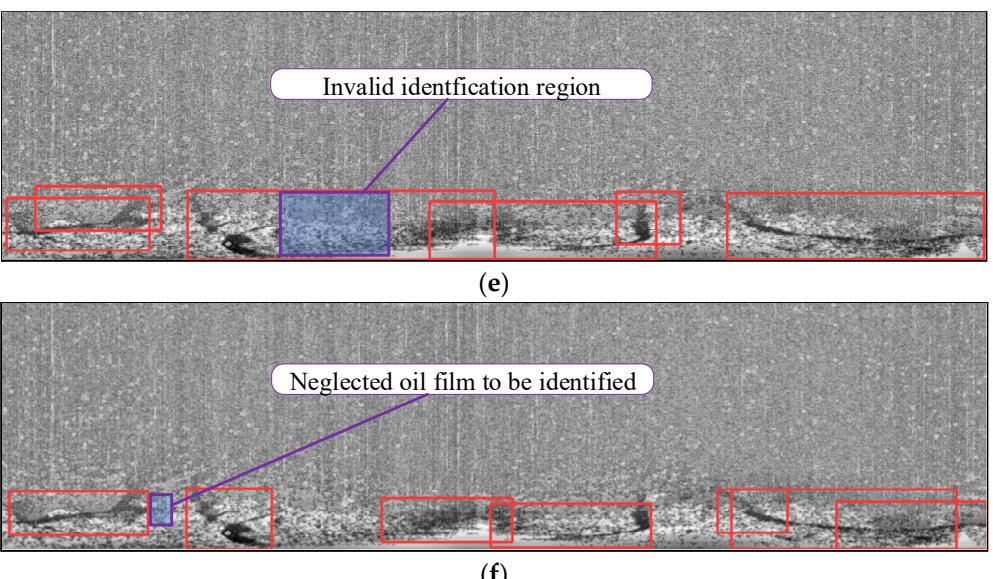

**Figure 19.** Oil film detection results of YOLO Deep Learning Network with different settings. (**a**) epochs: 100, batch size: 16, image size: 128 × 128; (**b**) epochs: 300, batch size: 16, image size: 128 × 128; (**c**) epochs: 200, batch size: 8, image size: 128 × 128; (**d**) epochs: 200, batch size: 32, image size: 128 × 128; (**e**) epochs: 200, batch size: 16, image size: 256 × 256; (**f**) epochs: 200, batch size: 16, image size: 128 × 128.

## 5. Conclusions

A marine radar oil spill detection method was put forward here with YOLO Deep Learning Network. The oil spill target identification influences of ship wake and long-range detection were analyzed. After data preprocessing, marine radar oil spill images were used to train the YOLOv5 detection model. Compared with previous methods, the detection efficiency of the method proposed here showed a greater advantage. In Current Stage, the deep learning method was only used to detect oil spill locations. The adaptive threshold was applied to finally segment the oil films. As for now, this method can be considered a preliminary prototype. With the accumulation of marine radar oil spill data under different sea conditions and the optimization of deep learning network, more applicable oil spill detection technologies will be realized. Oil film semantic segmentation with deep learning methods will also be developed in a future work.

**Author Contributions:** Conceptualization, B.L. and J.X.; methodology, B.L., J.X. and X.P.; software, R.C., L.M. and J.Y.; validation, L.C., J.L. and H.W.; formal analysis, B.L., Z.L. and Z.Z.; investigation, J.X., X.P. and R.C.; resources, J.X., L.M. and H.W.; data curation, H.W. and J.L.; writing—original draft preparation, B.L. and J.X.; writing—review, J.X. and X.P.; visualization, L.C.; supervision, J.Y.; project administration, X.P. and J.X.; funding acquisition, X.P. All authors have read and agreed to the published version of the manuscript.

**Funding:** This research was funded by the National Natural Science Foundation of China, grant numbers 52071090, 51879024; the Natural Science Foundation of Guangdong Province, grant number 2022A1515011603; the University Special Projects of Guangdong Province, grant number 2022ZDZX3005; the Natural Science Foundation of Shenzhen, grant number JCYJ20220530162200001; and the Research start-up funding project of Guangdong Ocean University, grant numbers 060302132009, 060302132106.

**Institutional Review Board Statement:** Not available.

**Informed Consent Statement:** Not available.

**Data Availability Statement:** The experimental shipborne radar images were collected by scholars from the Dalian Maritime University. The participants did not agree to share their data publicly.

**Acknowledgments:** The authors of this research would like to thank all the field management staff at the teaching–training ship *Yukun* of Dalian Maritime University during our research.

**Conflicts of Interest:** The authors declare no conflict of interest.

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
