# Peer review of "Preliminary Investigation on Marine Radar Oil Spill Monitoring Method Using YOLO Model"

_jmse, doi:10.3390/jmse11030670_

Round 1

Reviewer 1 Report

This is a typical work where researchers use deep learning techniques to solve detection and  classification problems without a  real understanding of the problem:

1.-The problem is clearly identified and presented.

2.-The algorithms to solve the problem are presented: pre-processing in figure 3 and the detection algorithm in figure 6. The authors do not analyse the structure of their algorithms, the influence of the different steps or the need of such steps. They train the network without asking themselves whether this is the optimal solution or whether they can reduce the complexity of the network.

3.-The size of the database for validating the algorithm is really small. The comparison with classical approaches is also reduced

Therefore, the algorithm could be potentially useful. However, this kind of approaches, which are very common nowadays, does not improve the knowledge of the problem.

Author Response

Dear editors and reviewers:

Thank you very much for your letter and the referees’ reports. Based on your comment and request, we have carefully revised the original manuscript. Thank you again for the suggestions put forward by the reviewers, which are of great help to improve the quality of our manuscript.

Here, we attached the revised manuscript in the formats of both PDF and editable words for your approval. A document answering every question from the referees was also summarized and enclosed.

A revised manuscript with the “Track Changes” version was attached as the supplemental material and for easy check and editing purpose. If you have any questions, please contact us without hesitation.

Reviewer 2 Report

Dear Authors,

The topic of the article is interesting and intelligible. The manuscript has certainly potential to improve. To help improve the quality of this manuscript, I have added more comments bellow:

Line:

-          27           „ecology“ is a science- rephrase the sentence

-          28           19-3?

-          30-31    rephrase- oil can leak from rig

-          35           4.27 major oil leak?

-          43-44    new energy function?

-          56           use of abbreviations that haven't been previously explained – the same in line 129 and 266)

-          58-68    should be the first paragraph of the Introduction chapter

-          78           space needed between Once and (YOLO)

-          84-87    rephrase (same in lines 125, 133-135, 251-253, 276 and 296-298)

-          90           as in Figure 1 (same in lines 102,

-          91           here you name the oil spill accident Dalian 7.16, but in the line 68 it is named the Dalian 716 oil spill accident

-          94           why changes font of the text in brackets?

-          96           title of the Figure should be: Marine radar remote sensing images of the Chinese Dalian 716 oil spill accident taken on (the date) at: a) 23:19:34; b) 23:21:24

-          98           title of the figure should be: The teaching-training ship Yukun used for marine radar images acquisition

-          99           the reference is missing- the performance characteristics of the marine radar device are given by the producer

-          101        method described in literature

-          105        if the marine radar image processing method described in literature 23 was used, then this should be stated in Figure’s 3 title as the reference

-          111        the title of the Fig. 4 should not be on the different page from the figure

-          114        lapsus calami- Markding

-          115        manually

-          117        rephrase- the sentence can’t start with the number

-          121        as shown in Figure 5 (same for lines 202, 203, 217, 219, 221, 234, 235, 239, and 255); after row 121 there should be a space before Fig. 5 (same for lines: 188, 204, 239, and 257)

-          124        The YOLO Model

-          134        This sentence is in contradiction to the first sentence of 2.4 subchapter (line 125-126)

-          145-147                explain why these parameters are used

-          163        space needed after “al.”

-          164        the manuscript should be written in indefinite form (avoid so-called “we-form”) (same for line 219)

-          167        the subtitle 2.6 should be on the next page

-          170        reference needed

-          173        was is

-          196        “Preliminary”-why capital letter?

-          210        films; (d);

Best regards,

Reviewer

Author Response

(The authors gave the same response as above.)

Reviewer 3 Report

You have a lot of spelling errors, such as menthod in line82 or 2.3. Image Markding. Check the entire paper.

Introduction:

- Please state the novelty and/or contributions of your paper clearly in the Introduction.

- I'm also interested in reading about your motivation to use YOLO instead of some other ANN for oil spills.

- Also, why to use ANN at all? Why not wavelets, Fourier or some transform?

- Is YOLO only a tool for your work or you made contribution in changing YOLO architecture?

Materials and Methods:

- Line 90 - Please, specify type of the radar systems and details which you can. You could connect this with Table 1 as an explanation of the table.

- Line 91 - Why have you decided to use 13 years old data?

- Figure 2 is useless. It shows some ship. It should be improved by adding important details or removed from the paper.

- Fig 3: Is ti "preprocessed images", not preprocessing? 

- Line 119: Did you mark or annotate or label the images? There are slight differences between terms.

- Line 137: Do you have some reference that Yolo is one of the most popular ANN?

Section 3.3:

- Explanation of Figure 9 is not satisfactory. You should describe what is in the images.

- Explanations about coordinate systems should be visible in the image or you should have a figure more that shows coordinates systems which are mentioned.

- Perhaps you should make some diagram which illustrates steps mentioned in the subsection.

References:

- 10/33 references are self-citations. You should reduce self-citations rate or by reducing these references or by increasing other references.

Author Response

(The authors gave the same response as above.)

Round 2

Reviewer 3 Report

You should explain in clear manner: what is the total number of images? What is the number of images in the training set? How did you choose them? What is the number of images in validation set? You should provide the statistics that prove that your number of images is sufficient. Otherwise, you should increase the number of images in both validation and training set. I'm aware that it is hard to find data. Perhaps you should consider using  augmentation methods of synthetically generated images to cope with this problem. As for now, this can be considered as preliminary findings.

When you revise Section 3.3, you forgot that Fig. 10 belongs to Discussion section. Furthermore it would look better if you choose something like: Block "" in Fig, 9 results in xxx in Fig. 10....

Author Response

Dear editors and reviewer:

Thank you very much for your letter and the reviewer’s report. Based on your comment and request, we have carefully revised the manuscript. Thank you very much for the guidance of the reviewer, which made the key content of our manuscript clearer.

Here, we attached the revised manuscript in the format of editable words for your approval. The responses answering every question from the reviewer was also summarized and enclosed. A revised manuscript with the “Track Changes” version was attached as the supplemental material and for easy check and editing purpose. If you have any questions, please contact us without hesitation.

Round 3

Reviewer 3 Report

The title should be revised in order to emphasis that this is preliminary research. Maybe including word like preliminary or case study or preliminary investigation.

Some statistical test of significance should be added. ANOVA, t or p test should be added.

Author Response

Dear editors and reviewer:
Thank you very much for your letter and the reviewer’s report. Based on your comment and request, we have carefully revised the manuscript. Thank you very much for the guidance of the reviewer, which make our manuscript more accurate.
Here, we attached the revised manuscript in the format of editable words for your approval. The responses answering every question from the reviewer was also summarized and enclosed. A revised manuscript with the “Track Changes” version was attached as the supplemental material and for easy check and editing purpose. If you have any questions, please contact us without hesitation.
